# Elevation of White Blood Cell Subtypes in Adult Trauma Patients with Stress-Induced Hyperglycemia

**DOI:** 10.3390/diagnostics13223451

**Published:** 2023-11-15

**Authors:** Cheng-Shyuan Rau, Spencer Chia-Hao Kuo, Ching-Hua Tsai, Sheng-En Chou, Wei-Ti Su, Shiun-Yuan Hsu, Ching-Hua Hsieh

**Affiliations:** 1Department of Neurosurgery, Kaohsiung Chang Gung Memorial Hospital and Chang Gung University, Kaohsiung City 83301, Taiwan; ersh2127@adm.cgmh.org.tw; 2Department of Plastic Surgery, Kaohsiung Chang Gung Memorial Hospital and Chang Gung University, Kaohsiung City 83301, Taiwan; spenc19900603@gmail.com; 3Department of Trauma Surgery, Kaohsiung Chang Gung Memorial Hospital and Chang Gung University, Kaohsiung City 83301, Taiwan; tsai1737@cloud.cgmh.org.tw (C.-H.T.); athenechou@gmail.com (S.-E.C.); s101132@adm.cgmh.org.tw (W.-T.S.); ah.lucy@hotmail.com (S.-Y.H.)

**Keywords:** trauma, monocyte, neutrophil, lymphocyte, platelet, mortality

## Abstract

Background: Blood immune cell subset alterations following trauma can indicate a patient′s immune–inflammatory status. This research explored the influence of stress-induced hyperglycemia (SIH) on platelet counts and white blood cell (WBC) subtypes, including the derived indices of the monocyte-to-lymphocyte ratio (MLR), neutrophil-to-lymphocyte ratio (NLR), and platelet-to-lymphocyte ratio (PLR), in trauma patients. Methods: We studied 15,480 adult trauma patients admitted from 1 January 1998 to 31 December 2022. They were categorized into four groups: nondiabetic normoglycemia (NDN, *n* = 11,602), diabetic normoglycemia (DN, *n* = 1750), SIH (*n* = 716), and diabetic hyperglycemia (DH, *n* = 1412). A propensity score-matched cohort was formed after adjusting for age, sex, and comorbidities, allowing for comparing the WBC subtypes and platelet counts. Results: Patients with SIH exhibited significantly increased counts of monocytes, neutrophils, and lymphocytes in contrast to NDN patients. However, no significant rise in platelet counts was noted in the SIH group. There were no observed increases in these cell counts in either the DN or DH groups. Conclusions: Our results demonstrated that trauma patients with SIH showed significantly higher counts of monocytes, neutrophils, and lymphocytes when compared to NDN patients, whereas the DN and DH groups remained unaffected. This underscores the profound association between SIH and elevated levels of specific WBC subtypes.

## 1. Introduction

For many years, there has been a recognized link between trauma and elevated blood sugar levels, known as hyperglycemia [1,2]. When individuals are hospitalized at medical facilities and exhibit symptoms of elevated blood glucose levels, their condition can be classified as either diabetic hyperglycemia (DH) or stress-induced hyperglycemia (SIH). The diagnosis of DH or SIH is established by observing serum glucose levels equal to or exceeding 200 mg/dL in patients with diabetes mellitus (DM) and individuals without DM, respectively [3].

Stressful situations often lead to SIH, especially in patients dealing with severe medical conditions like heart attacks [4,5,6], traumatic brain injuries [7,8,9], femoral fractures [10], or other significant severe traumas [11,12,13,14]. Notably, even those without any prior history of diabetes can experience elevated blood sugar levels, with around 25% of trauma patients who do not have a history of diabetes exhibiting hyperglycemia at the time of admission [15]. While DH is a long-term condition linked to changes in the small blood vessels due to extended periods of high blood sugar [16], SIH is a direct outcome of stress. In response to stress, the body releases hormones such as cortisol, epinephrine, norepinephrine, and glucagon. These hormones facilitate the release of glucose stored in the liver and reduce insulin production, resulting in a spike in blood sugar levels. Furthermore, SIH is marked by a decrease in insulin production, resistance to insulin in peripheral tissues, increased adrenal cortical activity, and a surge in inflammatory cytokines in the bloodstream [16,17].

The consequences of SIH are profound. It has been linked to a rise in both morbidity and mortality among trauma patients [16]. Additionally, it has been discovered that trauma patients with SIH have an increased mortality risk in comparison to those with DH [8,10,17,18,19,20,21,22,23,24]. Trauma patients with SIH have also been found to be at higher risk of experiencing trauma-related distress [25] for both acute and delayed adverse outcomes, which can impact their health-related quality of life [26].

Following a traumatic event, many significant time-dependent alterations in phenotypic characteristics have been seen in blood immune cell subsets, including monocytes, neutrophils, lymphocytes, and platelets. Subsequent to such injuries, the systemic inflammatory reaction elicits modifications in white blood cell (WBC) concentrations. There is an immediate surge in neutrophils post-trauma, instrumental in the acute inflammatory phase; however, a temporary decline may be observed owing to their adherence to endothelial linings or their navigation to injured locales. If there are ensuing complications, a sustained elevation might also be documented [27]. Concurrently, lymphocytes often manifest a pronounced reduction following trauma, a condition described as “trauma-induced lymphopenia”, though they tend to revert to baseline during convalescence [28,29,30]. The levels of monocytes may exhibit initial variability post-injury, but a subsequent augmentation is noted as they undergo differentiation and contribute to tissue recuperation [29,30,31]. Platelets, fundamental for coagulation, might experience an initial decrement, particularly in the context of pronounced hemorrhage, but they frequently rebound to bolster tissue restitution and maintain hemostasis [32]. 

The magnitude and nature of these cellular shifts can be contingent on the trauma’s character and intensity, underscoring the necessity for continual surveillance. Under conditions of physiological stress or pervasive inflammation, alterations in WBC subtype ratios can mirror patients′ immunological and inflammatory profiles, potentially prognosticating outcomes [33,34]. Moreover, derived indices such as the monocyte-to-lymphocyte ratio (MLR), neutrophil-to-lymphocyte ratio (NLR), and platelet-to-lymphocyte ratio (PLR) offer valuable perspectives on post-traumatic inflammatory and immunological dynamics. These ratios, notably NLR, MLR, and PLR, stand as facile, cost-efficient prognostic tools for trauma patient assessment [35,36,37,38,39,40,41].

Given that cell counts vary to reflect the host′s immune–inflammatory response, the aim of this study was to investigate the effects of SIH on platelets, WBC subtypes, and the derived ratios of NLR, MLR, and PLR in trauma patients. These metrics were examined between trauma patients who presented with SIH and patients who had nondiabetic normoglycemia (NDN). To ensure a comprehensive review, we simultaneously investigated other glucose-related conditions, including diabetic normoglycemia (DN) and DH.

## 2. Materials and Methods

### 2.1. Statement of Ethics

A retrospective cross-sectional analysis was undertaken utilizing the trauma registry system of Chang Gung Memorial Hospital. Permission for this study was given by the Institutional Review Board (IRB) of the hospital, with the permission codes 202100761B0 and 202100301B0. The research was granted an exemption from obtaining informed consent in accordance with the rules set forth by the IRB.

### 2.2. Inclusion and Grouping Criteria for Patients

In this research, adult trauma patients aged 20 years and above were examined, drawing from a pool of 50,310 trauma patients admitted to the hospital via the emergency room from 1 January 1998 to 31 December 2022 (as illustrated in Figure 1). The exclusion criteria included patients with specific trauma injuries such as burns, hanging injuries, or drowning, and those who lacked full admission data on glucose levels, specific WBC subsets, and platelet counts. This left us with a total of 15,480 patients for the analysis. For the purpose of this study, “Hyperglycemia” denotes emergency department serum glucose levels at or above 200 mg/dL. The study classified patients into four distinct groups: NDN, DN, SIH, and DH, based on the below definitions.

A DM diagnosis was made based on the American Diabetes Association′s guidelines, which involves referencing the patient’s history or detecting glycated hemoglobin (HbA1c) levels equal to or greater than 6.5% upon admission [3].“NDN” signifies serum glucose levels less than 200 mg/dL in individuals without a DM diagnosis.“Diabetic normoglycemia” or “DN” stands for serum glucose measurements below 200 mg/dL in DM-diagnosed patients.Patients with DM and serum glucose levels of 200 mg/dL or higher were identified as having “DH”, while those without DM diagnosis with similar glucose levels were labeled “SIH”.

### 2.3. Collection of Clinical Data

From the trauma registry system, various medical details were collected. These details encompassed patient demographics such as sex and age; blood cell counts covering monocytes, neutrophils, lymphocytes, and platelets; and a history of existing health conditions, including cerebrovascular accident (CVA), hypertension (HTN), coronary artery disease (CAD), congestive heart failure (CHF), diabetes mellitus (DM), and end-stage renal disease (ESRD). Additionally, metrics like the Glasgow Coma Scale (GCS) and Injury Severity Score (ISS) were taken into account. Upon patients’ entrance to the emergency room, laboratory data were also gathered. This data involved admission glucose levels, HbA1c, and specific subtypes of WBCs and platelets. Ratios such as the MLR, NLR, and PLR were computed by dividing the respective counts of monocytes, neutrophils, and platelets by the lymphocyte count. Notably, these cell counts were represented in concentrations of 10^6^/L. Lastly, records were kept regarding the duration of each patient′s stay in the hospital, measured in days, as well as any cases of in-hospital mortality.

### 2.4. Statistical Analyses

Analytical evaluations were conducted utilizing the SPSS 23.0 software designed for Windows, produced by IBM Corp., situated in Armonk, NY, USA. The Kolmogorov–Smirnov test was used to analyze the normalization of the distributed data for continuous variables. For continuous variables, an initial one-way analysis of variance was deployed, which was then followed up with Games–Howell post hoc tests. The results from these tests are presented as a mean with its corresponding standard deviation. To contrast categorical datasets, either Fisher’s exact tests or Pearson’s chi-squared (χ^2^) tests were applied. Moreover, odds ratios (ORs) were determined along with their 95% confidence intervals (CIs).

To ensure that the study sufficiently accounted for possible inherent discrepancies in baseline characteristics across different patient groups, especially when evaluating mortality outcomes, a specific method was employed. Utilizing the NCSS 10 software, originating from NCSS Statistical Software in Kaysville, Utah, a 1:1 propensity score-matched cohort was constructed. The methodology utilized was the Greedy technique, employing a caliper width of 0.2. The propensity scores in question were deduced through a logistic regression model which factored in certain covariates: sex, age, and previously identified health conditions. For discerning statistically significant variations between groups, a predetermined *p*-value threshold of less than 0.05 was established.

## 3. Results

### 3.1. Demographics and Patient Characteristics

Table 1 illustrates the distribution of the 15,480 participants across four categories: NDN (*n* = 11,602), DN (*n* = 1750), SIH (*n* = 716), and DH (*n* = 1412). In terms of sex, the SIH group′s distribution was similar to the NDN′s, whereas both the DN and DH categories displayed a more pronounced female representation. The patients with DN, SIH, and DH were on average older than the patients with NDN. When examining existing health conditions such as CVA, HTN, CAD, CHF, and ESRD, patients with DN or DH showed a noticeably higher prevalence compared to the NDN group. Yet, the SIH group’s medical history was not significantly different from the NDN group in terms of these conditions. Concerning the GCS score, the DN, SIH, and DH groups presented lower scores when compared with the NDN group. In terms of injury severity, the SIH group, with a median ISS of 15 (median {IQR: Q1–Q3}, 15 {7–15}), notably surpassed NDN’s 9 (IQR spanning from 4 to 13), DN’s 9 (IQR ranging from 4 to 14), and DH’s 9 (IQR from 8 to 16), with all these differences being statistically meaningful (*p* < 0.05). Lastly, the patients with DN and DH had greater injury severity compared to those in the NDN group.

### 3.2. Level of Subtypes of WBC and Platelets and the Derived Ratio

As presented in Table 1, the monocyte levels among patients with DN or DH were similar to those in the NDN group. However, the SIH group exhibited higher monocyte counts compared to the NDN group. Neutrophil levels were lower in the DN group than in the NDN group, whereas patients with SIH had elevated neutrophil counts relative to those with NDN. Interestingly, the neutrophil counts between the DH and NDN groups did not show a significant difference. When considering lymphocyte levels, there were notable differences between the DN, SIH, or DH groups and the NDN group. Specifically, lymphocyte counts in DN or DH patients were significantly diminished compared to the NDN group, whereas the SIH group had higher lymphocyte levels than the NDN patients. In terms of platelet counts, both the DN and DH groups had significantly lower levels compared to the NDN group. However, no significant difference in platelet levels was observed between the SIH and NDN groups. Lastly, for metrics like MLR, NLR, and PLR, no notable disparities were found when comparing patients with DN, SIH, or DH groups to those with NDN.

### 3.3. Outcomes of the Patients

Patients diagnosed with SIH and DH had longer hospital stays compared to the NDN group, whereas the duration of hospitalization for the DN group was similar to that of the NDN group. In terms of mortality rates, the DN, SIH, and DH groups recorded rates of 3.1%, 18.6%, and 4.8%, respectively, which were higher than the 2.1% observed in the NDN group. Specifically, the mortality rates for the DN, SIH, and DH groups were 1.48, 10.62, and 2.63 times higher, respectively, than that of the NDN group, as detailed in Table 1.

### 3.4. Level of Subtypes of WBC and Platelets in the Propensity Score-Matched Patient Cohorts

For comparisons between DN and NDN, SIH and NDN, DH and NDN, and SIH and DH, well-matched pairs of patients were created using propensity scores, as illustrated in Table 2, Table 3, Table 4 and Table 5 respectively. Among these matched cohorts, no significant discrepancies were noted in terms of sex, age, or comorbidities. When comparing the DN and NDN groups, and the DH and NDN groups, there were no significant differences in the counts of monocytes, neutrophils, lymphocytes, and platelets. However, as highlighted in Table 3, the SIH group had notably elevated counts of monocytes (594 ± 411 vs. 543 ± 338 {106/L}, *p* = 0.011), neutrophils (9704 ± 5484 vs. 8846 ± 4506 {106/L}, *p* = 0.001), and lymphocytes (2017 ± 1483 vs. 1756 ± 1081 {106/L}, *p* < 0.001) compared to the NDN group. The platelet levels, however, were not different between the SIH and NDN patients. Notably, when comparing the SIH and DH groups, there were no significant differences in the counts of monocytes, neutrophils, lymphocytes, and platelets (Table 5).

## 4. Discussion

This study revealed that the mortality rates for the DN, SIH, and DH groups were 1.48, 10.62, and 2.63 times higher, respectively, compared to the NDN group. Upon analyzing the platelets and subtypes of WBC in the propensity score-matched patient cohorts, it was observed that trauma patients with SIH had significantly higher numbers of monocytes, neutrophils, and lymphocytes compared to NDN patients. However, platelet counts were not elevated in the SIH group. No such elevations in monocytes, neutrophils, lymphocytes, or platelet counts were observed in DN or DH patients. Furthermore, despite the fact that some studies have implicated MLR, NLR, and PLR as useful biomarkers for trauma patients [35,38,42], this investigation found no significant differences between patients in the DN, SIH, or DH groups and those in the NDN group.

Trauma induces significant, time-sensitive alterations in the composition of blood immune cells, notably monocytes. These cells are pivotal in innate immunity and interact with antigen presentation to influence the adaptive immune response. They also facilitate tissue healing post-injury. For example, CD14+ monocytes in the bloodstream are believed to aid in vascular barrier restoration following trauma [43]. Inflammatory reactions post-trauma can impact the way monocytes differentiate in tissues, underscoring their adaptability and importance in the aftermath of injury [44]. A reduced expression of toll-like receptors (TLRs) and the human leukocyte antigen—DR isotype (HLA-DR) during acute inflammation might cause the observed post-trauma monocyte paralysis. This implies that trauma might diminish the monocytes’ capacity to detect and counter pathogens, potentially elevating infection risks [45].

Even though SIH patients exhibited more severe injuries, their monocyte counts were unexpectedly higher rather than lower. This seems to counter the notion that more severe trauma corresponds to monocyte deactivation, which is in line with the injury’s intensity and may result in these cells losing their antigen-presentation capabilities [46,47]. Nevertheless, elevated monocyte levels post-trauma have been linked to changes in cytokine responses [48]. Monocytes possess the ability to undergo differentiation into activated macrophages, facilitating the modulation of inflammatory reactions through the secretion of interleukin (IL)-10 and transforming growth factor-β1, ultimately expediting the process of tissue healing [49]. The role and implications of these elevated monocyte levels in SIH patients warrant further exploration.

Neutrophils are the predominant subtype of WBCs and serve as a frontline defense in the innate immune system. When activated, particularly during an intense acute inflammatory response, neutrophils can cause hyperinflammation and tissue damage. Increased concentrations of IL-1, tumor necrosis factor-α, and IL-6 are concurrently observed, alongside an uncontrolled outburst of polymorphonuclear cells and macrophages [50,51,52]. Following cortical trauma, there is a noted surge in neutrophils within the affected cortex [53]. For most trauma patients, neutrophils discharge web-like structures known as neutrophil extracellular traps, which have the capability to ensnare and eliminate pathogens [54]. Post-trauma, neutrophils are in an activated state, potentially elevating the chances of damage to the pulmonary microvasculature [55]. Changes in neutrophil behavior after trauma have been linked to the onset of complications such as acute respiratory distress syndrome and multiple organ failure [27]. The implications of increased neutrophil levels in SIH patients and their potential connection to post-traumatic complications merit deeper investigation.

Lymphocytes swiftly participate in both innate and adaptive immune reactions when faced with the stress of severe trauma [56]. Their immediate response is essential during the intense immune reaction that follows trauma and hemorrhagic shock [57]. A two-week study investigating lymphocyte behaviors post-severe trauma identified a marked activation in these cells, an activity typically unrelated to the patient’s clinical trajectory [58]. Post-trauma, T lymphocytes showed activation, leading to a noticeable decline in their proportion and overall count compared to a control group [58]. According to a prospective single-center study involving 130 patients, non-survivors had a greater lymphocyte count than survivors [59]. Moreover, research has demonstrated that there is an inverse correlation between ISS and the quantity of CD3+ T cells [60] following severe trauma. Lymphocytes decreased significantly in a sample of 105 trauma patients with an ISS greater than 20 during the first week following injury, with the greatest decline occurring on day three [61]. An estimated mortality rate of 45% has also been associated with patients presenting a lymphocyte count of ≤500 × 10^6^/L within the first 48 h post-injury [62]. Notably, this research found that even with the severe injuries sustained by SIH patients, their lymphocyte levels surpassed those in the control group. Comprehending the shifts in lymphocyte behaviors, both in their reduced responsiveness and heightened activation, offers valuable insights into a patient′s immune health and potential subsequent issues [63].

Platelet dysfunction is a frequent consequence of significant trauma [64,65,66]. Multiple studies have delved into the correlation between platelet counts and traumatic injuries. Following a severe injury, platelets exhibit a decreased responsiveness during ongoing bleeding [67]. In this research, there was no marked difference in platelet levels among patients with DN, SIH, or DH compared to those with NDN. It is important to underscore that even when platelet counts appear normal and clotting studies are standard, impactful platelet dysfunction can emerge post-trauma. Platelets play a crucial role in both the innate and adaptive immune responses, exerting a considerable impact on inflammation through their interactions with monocytes, neutrophils, lymphocytes, and the endothelium layer [68]. Such platelet dysfunction can have severe ramifications for patient survival [69].

As additional information about the causes of SIH that lead to immunological dysfunction after trauma is uncovered, tailored treatments to mitigate the long-term effects of trauma may be created [70]. However, further work is required to attain that goal. Furthermore, it has to be noted that this research comes with several limitations. Firstly, the retrospective design, coupled with the exclusion of incomplete data, could introduce selection bias. Notably, patients pronounced dead upon arrival at the emergency room were not captured in the trauma database, with only in-hospital deaths being recorded, potentially skewing the data analysis. Secondly, interventions such as blood transfusions or fluid challenges can affect platelet and WBC subtype levels, introducing outcome measurement biases. Thirdly, although invasive procedures and surgeries might influence patient outcomes, it is inferred that treatment approaches yielded consistent results among the participants. Additionally, this investigation does not delve into the intricacies of the mechanisms causing lymphocyte fluctuations. Lastly, our findings, stemming from a singular urban trauma center, might not extend applicability across diverse geographical settings.

## 5. Conclusions

The findings of this study indicated that trauma patients who presented with SIH exhibited significantly higher counts of monocytes, neutrophils, and lymphocytes compared to NDN patients, while those who had DN or DH did not. The results highlight the significant connection between SIH and the level of subtypes of WBCs. As we advance in medical knowledge, these insights can pave the way for improved diagnostic tools and treatment strategies in trauma care.

## Figures and Tables

**Figure 1 diagnostics-13-03451-f001:**
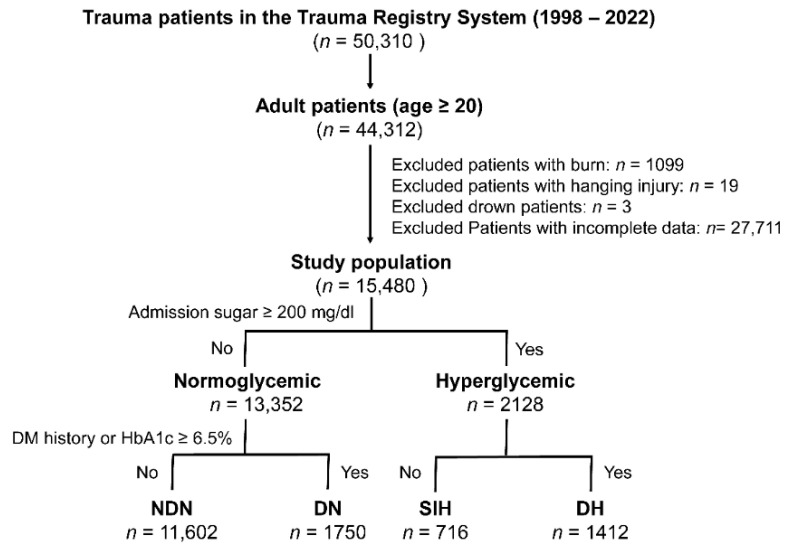
A flowchart depicting the process of categorizing the included adult trauma patients into four distinct groups.

**Table 1 diagnostics-13-03451-t001:** Comparison of demographics and patient characteristics of the four distinct groups of patients.

Variables	NDN *n* = 11,602	DN *n* = 1750	SIH *n* = 716	DH *n* = 1412	DN vs. NDN	SIH vs. NDN	DH vs. NDN
					*p*	*p*	*p*
Sex					<0.001	0.630	<0.001
Male, *n* (%)	6553 (56.5)	816 (46.6)	411 (57.4)	644 (45.6)			
Female, *n* (%)	5049 (43.5)	934 (53.4)	305 (42.6)	768 (54.4)			
Age, years	54.0 ± 19.5	69.2 ± 11.9	56.7 ± 18.3	67.0 ± 12.5	<0.001	<0.001	<0.001
Co-morbidities							
CVA, *n* (%)	377 (3.2)	190 (10.9)	21 (2.9)	130 (9.2)	<0.001	0.642	<0.001
HTN, *n* (%)	2897 (25.0)	1259 (71.9)	191 (26.7)	890 (63.0)	<0.001	0.307	<0.001
CAD, *n* (%)	382 (3.3)	215 (12.3)	24 (3.4)	150 (10.6)	<0.001	0.931	<0.001
CHF, *n* (%)	66 (0.6)	29 (1.7)	2 (0.3)	31 (2.2)	<0.001	0.310	<0.001
ESRD, *n* (%)	158 (1.4)	107 (6.1)	8 (1.1)	91 (6.4)	<0.001	0.582	<0.001
GCS, median (IQR)	15 (15–15)	15 (15–15)	15 (7–15)	15 (15–15)	0.001	<0.001	<0.001
ISS, median (IQR)	9 (4–13)	9 (4–14)	15 (7–15)	9 (8–16)	0.004	<0.001	<0.001
<16, *n* (%)	9049 (78.0)	1320 (75.4)	356 (49.7)	979 (69.3)	0.016	<0.001	<0.001
16–24, *n* (%)	1830 (15.8)	344 (19.7)	135 (18.9)	294 (20.8)	<0.001	0.029	<0.001
≥25, *n* (%)	723 (6.2)	86 (4.9)	225 (31.4)	139 (9.8)	0.031	<0.001	<0.001
Monocytes (count/μL)	547 ± 314	535 ± 302	598 ± 411	547 ± 338	0.143	<0.001	0.975
Neutrophils (count/μL)	8801 ± 4452	8293 ± 4475	9704 ± 5484	8784 ± 4296	<0.001	<0.001	0.892
Lymphocytes (count/μL)	1757 ± 1043	1626 ± 950	2017 ± 1483	1666 ± 1053	<0.001	<0.001	0.002
Platelets (count/μL)	2,251,163 ± 72,944	216,457 ± 70,588	2,231,905 ± 105,612	216,205 ± 66,483	<0.001	0.664	<0.001
MLR	0.4 ± 0.5	0.4 ± 0.4	0.4 ± 0.4	0.4 ± 0.4	0.343	0.841	0.155
NLR	7.1 ± 7.3	7.1 ± 9.1	7.6 ± 8.9	7.5 ± 7.0	0.949	0.095	0.059
PLR	165.3 ± 123.4	169.9 ± 157.4	165.5 ± 290.3	167.8 ± 108.6	0.165	0.978	0.477
Hospital stay (days)	9.2 ± 9.7	9.9 ± 10.9	14.9 ± 16.5	12.1 ± 12.7	0.756	<0.001	<0.001
Mortality, *n* (%)	244 (2.1)	54 (3.1)	133 (18.6)	68(4.8)	0.009	<0.001	<0.001
Mortality, OR (95%CI)					1.48 (1.10–2.00)	10.62 (8.46–13.33)	2.36 (1.79–3.10)

CAD, coronary artery disease; CHF, congestive heart failure; CI, confidence interval; CVA, cerebral vascular accident; DH, diabetic hyperglycemia; DM, diabetes mellitus; DN, diabetic normoglycemia; ESRD, end-stage renal disease; GCS, Glasgow Coma Scale; HTN, hypertension; ICU, intensive care unit; IQR, interquartile range; ISS, injury severity score; MLR, monocyte-to-lymphocyte ratio; NDN, non-diabetic normoglycemia; NLR, neutrophil-to-lymphocyte ratio; OR, odds ratio; PLR, platelet-to-lymphocyte ratio; and SIH, stress-induced hyperglycemia.

**Table 2 diagnostics-13-03451-t002:** The propensity scores-matched patient cohort was created to assess the differences in platelet counts and white blood cell (WBC) subtypes in the patients with diabetic normoglycemia (DN) vs. those with nondiabetic normoglycemia (NDN).

Propensity Score Matched—Patient Cohort								
Adjusted Variables	DN *n* = 1700		NDN *n* = 1700		OR (95%CI)		*p*	Standardized Difference
Male, *n* (%)	787	(46.3)	787	(46.3)	1.00	(0.87–1.14)	1.000	0.00%
Age, years	69.3	±12.0	69.3	±12.0	-		0.974	0.11%
CVA, *n* (%)	160	(9.4)	160	(9.4)	1.00	(0.79–1.26)	1.000	0.00%
HTN, *n* (%)	1216	(71.5)	1216	(71.5)	1.00	(0.86–1.16)	1.000	0.00%
CAD, *n* (%)	188	(11.1)	188	(11.1)	1.00	(0.81–1.24)	1.000	0.00%
CHF, *n* (%)	18	(1.1)	18	(1.1)	1.00	(0.52–1.93)	1.000	0.00%
ESRD, *n* (%)	78	(4.6)	78	(4.6)	1.00	(0.73–1.38)	1.000	0.00%
Outcome variables								
Monocytes (count/μL)	536	±304	527	±286	-		0.367	-
Neutrophils (count/μL)	8296	±4501	8464	±4106	-		0.253	-
Lymphocytes (count/μL)	1633	±955	1671	±1001	-		0.253	-
Platelets (count/μL)	216,683	±70,888	218,397	±71,777	-		0.484	-

CAD, coronary artery disease; CHF, congestive heart failure; CI, confidence interval; CVA, cerebral vascular accident; DN, diabetic normoglycemia; ESRD, end-stage renal disease; HTN, hypertension; NDN, non-diabetic normoglycemia; and OR, odds ratio.

**Table 3 diagnostics-13-03451-t003:** The propensity scores-matched patient cohort was created to assess the differences in platelet counts and white blood cell (WBC) subtypes in the patients with stress-induced hyperglycemia (SIH) vs. those with nondiabetic normoglycemia (NDN).

Propensity Score Matched—Patient Cohort								
Adjusted Variables	SIH *n* = 716		NDN *n* = 716		OR (95%CI)		*p*	Standardized Difference
Male, *n* (%)	411	(57.4)	411	(57.4)	1.00	(0.81–1.23)	1.000	0.00%
Age, years	56.7	±18.3	56.7	±18.3	-		0.990	0.07%
CVA, *n* (%)	21	(2.9)	21	(2.9)	1.00	(0.54–1.85)	1.000	0.00%
HTN, *n* (%)	191	(26.7)	197	(26.7)	1.00	(0.79–1.26)	1.000	0.00%
CAD, *n* (%)	24	(3.4)	24	(3.4)	1.00	(0.56–1.78)	1.000	0.00%
CHF, *n* (%)	2	(0.3)	2	(0.3)	1.00	(0.14–7.12)	1.000	0.00%
ESRD, *n* (%)	8	(1.1)	8	(1.1)	1.00	(0.37–2.68)	1.000	0.00%
Outcome variables								
Monocytes (count/μL)	594	±411	543	±338	-		0.011	-
Neutrophils (count/μL)	9704	±5484	8846	±4506	-		0.001	-
Lymphocytes (count/μL)	2017	±1483	1756	±1081	-		<0.001	-
Platelets (count/μL)	223,905	±105,612	220,062	±70,281	-		0.418	-

CAD, coronary artery disease; CHF, congestive heart failure; CI, confidence interval; CVA, cerebral vascular accident; ESRD, end-stage renal disease; HTN, hypertension; NDN, non-diabetic normoglycemia; OR, odds ratio; and SIH, stress-induced hyperglycemia.

**Table 4 diagnostics-13-03451-t004:** The propensity scores-matched patient cohort was created to assess the differences in platelet counts and white blood cell (WBC) subtypes in the patients with diabetic hyperglycemia (DH) vs. those with nondiabetic normoglycemia (NDN).

Propensity Score Matched—Patient Cohort								
Adjusted Variables	DH *n* = 1376		NDN *n* = 1376		OR (95%CI)		*p*	Standardized Difference
Male, *n* (%)	629	(45.7)	629	(45.7)	1.00	(0.86–1.16)	1.000	0.00%
Age, years	66.9	±12.6	66.9	±12.6	-		0.978	−0.10%
CVA, *n* (%)	119	(8.6)	119	(8.6)	1.00	(0.77–1.31)	1.000	0.00%
HTN, *n* (%)	858	(62.4)	858	(62.4)	1.00	(0.86–1.17)	1.000	0.00%
CAD, *n* (%)	125	(9.1)	125	(9.1)	1.00	(0.77–1.30)	1.000	0.00%
CHF, *n* (%)	18	(1.3)	18	(1.3)	1.00	(0.52–1.93)	1.000	0.00%
ESRD, *n* (%)	68	(4.9)	68	(4.9)	1.00	(0.71–1.41)	1.000	0.00%
Outcome variables								
Monocytes (count/μL)	546	±338	532	±291	-		0.237	-
Neutrophils (count/μL)	8785	±4308	8672	±4382	-		0.494	-
Lymphocytes (count/μL)	1671	±1063	1686	±983	-		0.707	-
Platelets (count/μL)	215,866	±66,325	216,465	±70,705	-		0.819	-

CAD, coronary artery disease; CHF, congestive heart failure; CI, confidence interval; CVA, cerebral vascular accident; DH, diabetic hyperglycemia; ESRD, end-stage renal disease; HTN, hypertension; NDN, non-diabetic normoglycemia; and OR, odds ratio.

**Table 5 diagnostics-13-03451-t005:** The propensity scores-matched patient cohort was created to assess the differences in platelet counts and white blood cell (WBC) subtypes in the patients with stress-induced hyperglycemia (SIH) vs. those with diabetic hyperglycemia (DH).

Propensity Score Matched—Patient Cohort								
Adjusted Variables	SIH *n* = 119		DH *n* = 119		OR (95%CI)		*p*	Standardized Difference
Male, *n* (%)	74	(62.2)	74	(62.2)	1.00	(0.59–1.69)	1.000	0.00%
Age, years	60.8	±14.1	60.9	±14.1	-		0.967	−0.54%
CVA, *n* (%)	4	(3.4)	4	(3.4)	1.00	(0.24–4.10)	1.000	0.00%
HTN, *n* (%)	34	(28.6)	34	(28.6)	1.00	(0.57–1.76)	1.000	0.00%
CAD, *n* (%)	3	(2.5)	3	(2.5)	1.00	(0.20–5.06)	1.000	0.00%
CHF, *n* (%)	0	(0.0)	0	(0.0)	-		-	0.00%
ESRD, *n* (%)	0	(0.0)	0	(0.0)	-		-	0.00%
Outcome variables								
Monocytes (count/μL)	583	±451	543	±403	-		0.468	-
Neutrophils (count/μL)	9658	±5188	9110	±4704	-		0.395	-
Lymphocytes (count/μL)	1870	±1178	1837	±1047	-		0.823	-
Platelets (count/μL)	244,479	±204,921	220,647	±74,245	-		0.234	-

CAD, coronary artery disease; CHF, congestive heart failure; CI, confidence interval; CVA, cerebral vascular accident; DH, diabetic hyperglycemia; DN, diabetic normoglycemia; ESRD, end-stage renal disease; HTN, hypertension; OR, odds ratio; and SIH, stress-induced hyperglycemia.

## Data Availability

Data is unavailable due to privacy and ethical restrictions.

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
