# Peer review of "Elevation of White Blood Cell Subtypes in Adult Trauma Patients with Stress-Induced Hyperglycemia"

_diagnostics, 2023, doi:10.3390/diagnostics13223451_

Round 1
Reviewer 1 Report
Comments and Suggestions for Authors
This is an article by Cheng-Shyuan Rau et al.
I would like to address several suggestions to authors that may improve the manuscript.
General recommendations
Please check all the text for spelling mistakes.
Page 1, line 34
Please delete “a”
Page 2 line 83-85
Please rephrase this sentence: “Under the hypothesis that the cell counts might alter to reflect the immune-inflammatory response of the host, this study aimed to investigate the implications of SIH on trauma patients' WBC subtypes, platelets, and the derived ratios of NLR, MLR, and PLR”
It is not hypothesis that blood cell counts reflect to inflammatory response. It is well documented data.
Please rewrite the aim of this study.
Page 3
According to the Figure 1, manny patients were excluded. Unfortunately, excluding criteria were not reported in the text.
Please write all excluding criteria in the text after section 2.2. Inclusion and Grouping Criteria for Patients.
Authors has not performed analysis between diabetic hyperglycemia (DH) and Stress-induced hyperglycemia (SIH). Please add this analysis as separate table
Author Response
Dear Reviewer 1
Thank you for your time, effort, and professional comments in regard to our manuscript entitled “Elevation of white blood cells subtypes in adult trauma patients with stress-induced hyperglycemia” to Diagnostics. I have done the revision of the document according to your recommendation and highlighted those areas in red color.
Comments:
I would like to address several suggestions to authors that may improve the manuscript.
General recommendations
Please check all the text for spelling mistakes.
Page 1, line 34, Please delete “a”
Answer: Yes, it is deleted.
Page 2 line 83-85, Please rephrase this sentence: “Under the hypothesis that the cell counts might alter to reflect the immune-inflammatory response of the host, this study aimed to investigate the implications of SIH on trauma patients' WBC subtypes, platelets, and the derived ratios of NLR, MLR, and PLR” It is not hypothesis that blood cell counts reflect to inflammatory response. It is well documented data. Please rewrite the aim of this study.
Answer: Yes, we had changed the description into “Given that cell counts vary to reflect the host's immune-inflammatory response, the aim of this study was to investigate at the effects of SIH on platelets, WBC subtypes, and derived ratios of NLR, MLR, and PLR in trauma patients.” (Lines 83-85) I hope that would make the statement better. Thank you.
Page 3
According to the Figure 1, many patients were excluded. Unfortunately, excluding criteria were not reported in the text. Please write all excluding criteria in the text after section 2.2. Inclusion and Grouping Criteria for Patients.
Answer: The exclusion criteria were stated in the section 2.2. of the revised article as “The exclusion criteria included those patients with specific trauma injuries such as burns, hanging injuries, or drowning, and those who lacked full admission data on glucose levels, specific WBC subsets, and platelet counts.” (Lines 99-102) Thank you.
Authors has not performed analysis between diabetic hyperglycemia (DH) and Stress-induced hyperglycemia (SIH). Please add this analysis as separate table
Answer: OK, the comparison between the SIH vs. DH was performed in a separate table and listed as the Table 5 in the revised manuscript. The corresponding description was made in the results and discussion sections (Lines 193, 202-204, 231-236).
This article had revised under your kind suggestion, and we hope that will satisfy your standard. If required, we are very delighted to make further change or revision. Thank you
Ching-Hua Hsieh, M.D. Ph.D., FACS
Department of Plastic and Reconstructive Surgery, Kaohsiung Chang Gung Memorial Hospital, Chang Gung University College of Medicine, Taiwan.
Reviewer 2 Report
Comments and Suggestions for Authors
The authors present in the manuscript a study evaluating white blood cell subtypes in adult trauma patients with stress-induced hyperglycemia (SIH). They report that patients with SIH had significantly greater counts of monocytes, neutrophils, and lymphocytes compared with those who had nondiabetic normoglycemia. These findings may help improve the diagnosis and treatment in trauma care. Overall, the work is well done.
Minor comments
11) In view of all data presented in the manuscript, I assume the authors did data normality analysis. Please add this information to the subsection of “Statistical Analyses”.
22) In Tables 1-3, values of some parameters are not appropriately aligned, e.g., platelet numbers and OR of mortality in Table 1.
33) Grammatical errors, such as “did not significantly be different” (line 162).
Comments on the Quality of English LanguageMinor English editing is required.
Author Response
Dear Reviewer 2
Thank you for your time, effort, and professional comments in regard to our manuscript entitled “Elevation of white blood cells subtypes in adult trauma patients with stress-induced hyperglycemia” to Diagnostics. I have done the revision of the document according to your recommendation and highlighted those areas in red color.
The authors present in the manuscript a study evaluating white blood cell subtypes in adult trauma patients with stress-induced hyperglycemia (SIH). They report that patients with SIH had significantly greater counts of monocytes, neutrophils, and lymphocytes compared with those who had nondiabetic normoglycemia. These findings may help improve the diagnosis and treatment in trauma care. Overall, the work is well done.
Minor comments
In view of all data presented in the manuscript, I assume the authors did data normality analysis. Please add this information to the subsection of “Statistical Analyses”.
Answer: Yes, the normalization of the data is routinely checked before the use of different types of analysis. I had indicated in the section of statistical analysis with “The Kolmogorov–Smirnov test was used to analyze the normalization of the distributed data for continuous variables.” (Lines 137-138)
In Tables 1-3, values of some parameters are not appropriately aligned, e.g., platelet numbers and OR of mortality in Table 1.
Answer: Sorry for the disarrangement of the parameters in the Tables due to the editing work by Diagnostics. I had reformed them entirely to make a better presentation. Thank you.
Grammatical errors, such as “did not significantly be different” (line 162).
Answer: Yes, the grammatical errors were corrected as “was not significantly different from” (Lines 161)
Ching-Hua Hsieh, M.D. Ph.D., FACS
Department of Plastic and Reconstructive Surgery, Kaohsiung Chang Gung Memorial Hospital, Chang Gung University College of Medicine, Taiwan.